# Target pollen isolation using automated infrared laser-mediated cell disruption

Ikuma Kaneshiro[1,2], Masako Igarashi[1], Tetsuya Higashiyama[1,3,4], and Yoko Mizuta[1,5] 

[1]Institute of Transformative Bio-Molecules (WPI-ITbM), Nagoya University, Furo-cho, Chikusa-ku, Nagoya, Aichi 464-8601, Japan; [2]Department of Physics, Graduate School of Science, Nagoya University, Furo-cho, Chikusa-ku, Nagoya, Aichi 464-8602, Japan; [3]Division of Biological Sciences, Graduate School of Science, Nagoya University, Furo-cho, Chikusa-ku, Nagoya, Aichi 464-8602, Japan; [4]Department of Biological Sciences, Graduate School of Science, The University of Tokyo, 7-3-1 Hongo, Bukyo-ku, Tokyo113-0033, Japan; [5]Institute for Advanced Research (IAR), Nagoya University, Furo-cho, Chikusa-ku, Nagoya, Aichi 464-8601, Japan

## Original Research Article

**Keywords:**
cell disruption; laser irradiation; biolistic delivery; pollen; *Nicotiana*.

**Author for correspondence:**
Y. Mizuta,
E-mail: mizuta.yoko.u6@f.mail.nagoya-u.ac.jp

### Abstract

Single-cell analysis is important to understand how individual cells work and respond at the cell population level. Experimental single-cell isolation techniques, including dilution, fluorescence-activated cell sorting, microfluidics, and micromanipulation, have been developed in recent decades. However, such applications typically require large cell populations and skilled professionals. Additionally, these methods are unsuitable for sequential analysis before and after cell isolation. In this study, we propose a method for target cell isolation using automated infrared laser-mediated disruption of pollen grains in pollen populations. Germination of the target pollen was observed at the same location as that before laser irradiation, and germinated pollen grains were enriched in the cell population. Pollination of laser-irradiated bulk pollen populations also showed that the target pollen preferentially germinated on the stigma. This method is expected to facilitate physiological analyses of target cells at the single-cell level and effectively produce seeds derived from target pollen.

## 1. Introduction

A cell is a fundamental unit in living organisms. In cell biology, analysing the selection of specific target cells apart from other cells is necessary to gain a better understanding of their biological characteristics such as gene function, cell structure, and development. Methods used to separate target cells from other cells promote the study of biology, physiology, biochemistry, and taxonomy at the single-cell level (Gross et al., 2015). Target cell isolation techniques can be divided into two major categories: (1) selection of targeted cells and (2) elimination of non-target cells. Single-cell isolation and sorting techniques have been refined in recent decades (Hu et al., 2016). Fluorescence-activated cell sorting and microfluidic droplet-based platforms have been widely used for high-throughput isolation and selection of targeted cells; however, these require a large number of cells, which interferes with cell location and phenotypic profile. Micromanipulation, which involves manual cell picking with micropipettes using a microscope, is a simple approach for single-cell isolation; however, its throughput is limited and requires skilled professionals. In the latter category of isolation techniques, the most commonly known method of eliminating non-target cells is the selection of *Escherichia coli* using antibiotics. Traditionally, antibiotics and antibiotic-resistance genes have been used to select *E. coli* strains harbouring recombinant plasmids. However, some cell species are insensitive or resistant to antibiotics (Alanis, 2005). Therefore, a means of isolating target cells without relying on conventional methods is required.

Pollen grains are male gametophytes in flowering plants. Pollen grains can be easily collected from anthers as isolated cells and can be conveniently used for experiments as a uniform isolated cell population. Biolistic delivery, also referred to as particle bombardment, is widely used to transiently introduce and express exogenous genes in cells (Sanford, 2000). Recently, we reported a method for the biolistic delivery of genes into the pollen of *Nicotiana benthamiana*, which is easy to handle and has abundant pollen grains (Nagahara et al., 2021). However, the efficiency of gene introduction into pollen by bombardment is low, and pollen isolation using antibiotic-resistance genes cannot be used (Hoffmann et al., 1988). Therefore, a method for

isolating gene-introduced pollen for efficient observation and pollination is required.

Herein, we propose a method for target cell isolation by the selective disruption of cells in a specific medium using infrared (IR) laser irradiation of *N. benthamiana* pollen. Target pollen grains were transiently labelled by the expression of fluorescent proteins that were mechanically extracted from the pollen population. Automatic enrichment of labelled cells is possible with this method. In addition, this method facilitates the enrichment of target cells while maintaining cell location and phenotypic profile, such as pollen germination in the culture medium. It also allows pollination of the extracted pollen grains, which suggests that seeds may be produced efficiently.

## 2. Methods

### 2.1. Plant materials and growth conditions

In this study, wild-type and transgenic *N. benthamiana*, *AtUBQ10p::H2B-mClover*, *35Sp::H2B-tdTomato*, *AtUBQ10p::tdTomato*, and wild-type *Nicotiana tabacum* 'Petit Havana SR1' described in a previous study were used (Nagahara et al., 2021). Seeds were sterilised for 10 min using a solution containing 1% (v/v) sodium hypochlorite (Fujifilm Wako Pure Chemical, Osaka, Japan) and 0.02% (v/v) Triton X-100 (Nacalai Tesque, Kyoto, Japan). After washing with sterilised water, the seeds were sown on a medium containing 1× Murashige and Skoog Basal Medium (#M0404; MERCK, Darmstadt, Germany), 3% (w/v) sucrose (Fujifilm Wako Pure Chemical), and 0.8% (w/v) Bacto Agar (Thermo Fisher Scientific, Waltham, MA, USA) adjusted to pH 5.8 with KOH. Plants were germinated and grown in a growth chamber at 25–30°C after cold treatment at 4°C for 2–3 d. Two-week-old seedlings were transferred to the soil and grown in a greenhouse at 25–30°C under long-day conditions (16 hr light/8 hr dark).

### 2.2. Plasmid construction

The plasmid vector *AtUBQ10p::H2B-tdTomato* (MUv2458) used for particle bombardment was provided by Dr. Minako Ueda (Tohoku University). The fluorescent protein tdTomato was fused with *Arabidopsis thaliana H2B* (*HISTONE 2 B*; *At1g07790*), which was driven by the *A. thaliana UBQ10* (*UBIQUITIN 10*; *At4g05320*) promoter (634 bp). Each fragment was cloned into the pGreen0029 vector (Hellens et al., 2000).

### 2.3. Pollen number counting

A single anther from wild-type *N. benthamiana* was suspended in 100 μL pollen culture medium. The pollen culture medium had the following composition: 0.5 M sucrose, 3 mM glutamine, 10 g/L lactalbumin hydrolysate, 10 mM $KNO_3$, 1 mM Ca $(NO_3)_2 \cdot 4H_2O$, 1 mM $MgSO_4 \cdot 7 H_2O$, 0.16 mM $H_3BO_3$, 1 mM uridine, 0.5 mM cytidine, and 1 mM phosphate buffer (pH 7.0) (Tupý et al., 1991). Of these, 10 μL of the pollen suspension was used for pollen counting using a cell counter (TC20; Bio-Rad Laboratories, Hercules, CA, USA) to calculate pollen density. Two anthers each from five flowers were analysed to estimate the average pollen number in a single anther.

### 2.4. Biolistic delivery of plasmid DNA into pollen

Pollen grains were used for biolistic delivery of plasmid vectors into cells. Pollen grains were obtained from buds 2 to 4 d before anthesis to analyse laser conditions and from just flowering flowers to analyse pollen germination and pollination. Pollen at 2–4 d before anthesis is suitable for analysing laser conditions for cell isolation because it cannot germinate pollen tubes. One-sixth to one-half of anthers were used in each experiment. Fresh anthers were squashed onto a wet filter with liquid pollen culture medium to disperse and absorb pollen. Biolistic delivery was performed as previously described (PDS-1000/He; Bio-Rad Laboratories; Nagahara et al., 2021). After bombardment, 1,000 μL of liquid pollen culture medium was added to the filter to suspend the pollen. The pollen suspension was then transferred to a 35-mm glass base dish (3910-035; Iwaki, AGC Techno Glass, Shizuoka, Japan) and cultured overnight in a chamber at 25°C. Gene introduction was confirmed based on the expression of fluorescent proteins (Figure 1a). Red fluorescent protein expression signals derived from the introduced plasmid DNA were observed using an inverted fluorescence microscope (emission filter: Cy3 570–610, Eclipse Ti2-E; Nikon, Tokyo, Japan) 18 hr after bombardment.

### 2.5. Visualisation of viable pollen using fluorescein diacetate (FDA) solution

After confirming fluorescent protein expression, a fluorochromatic test, also known as FDA staining, was used to assess cell viability (Heslop-Harrison et al., 1984). A 1% (w/v) solution of FDA (Merck) in acetone was added to the pollen suspension in the culture medium (Figure 1a) at a final concentration 0.019 μM at 18 hr after bombardment. After more than 10 min, cell viability was assessed based on green fluorescence signals using an inverted fluorescence microscope (emission filter: FITC 510–560, Eclipse Ti2-E; Nikon).

### 2.6. Infrared laser-evoked gene operator (IR-LEGO) microscope system and image analysis

An optical IR-LEGO system was used for the cell disruption of non-target pollen. An Eclipse Ti2-E microscope (Nikon) with an IR-LEGO-490 mini/E system (Sigma-Koki, Saitama, Japan) was used. An objective lens (CFI PlanApo λ 20×, NA = 0.75, WD = 1.0; Nikon) was used for IR laser focusing, irradiation of the target cells with the IR laser, and imaging of the samples. When the power of the IR laser source was set to 300 mW, the laser power delivered to the samples in the focal plane was 145 mW, as measured using a power meter (Coherent, Santa Clara, CA, USA). The irradiation exposure time for cell disruption of each pollen was set from 1 to 10 ms. Images were captured using a scientific CMOS camera (ORCA-Fusion BT C15440-20UP; Hamamatsu Photonics, Shizuoka, Japan). Bandpass filters used for autofluorescence of pollen grains included fluorescein isothiocyanate (FITC) for the FDA signal, Cy3 for tdTomato fluorescence, and cyan fluorescent protein (CFP) (Nikon). ImageJ software version 1.53j (https://imagej.nih.gov/ij/index.html) was used to generate and analyse the images.

### 2.7. Automated cell disruption for target cell isolation using a programmable software

The protocols for automated image acquisition, target cell extraction, and laser irradiation routines for target cell isolation were produced using NIS-Elements JOBS Programmable Software (Nikon). NIS-Elements JOBS was used as a plug-in module for NIS-Elements AR (Nikon), which enabled the design of experiments with customised acquisition and analyses. In this study, a five-

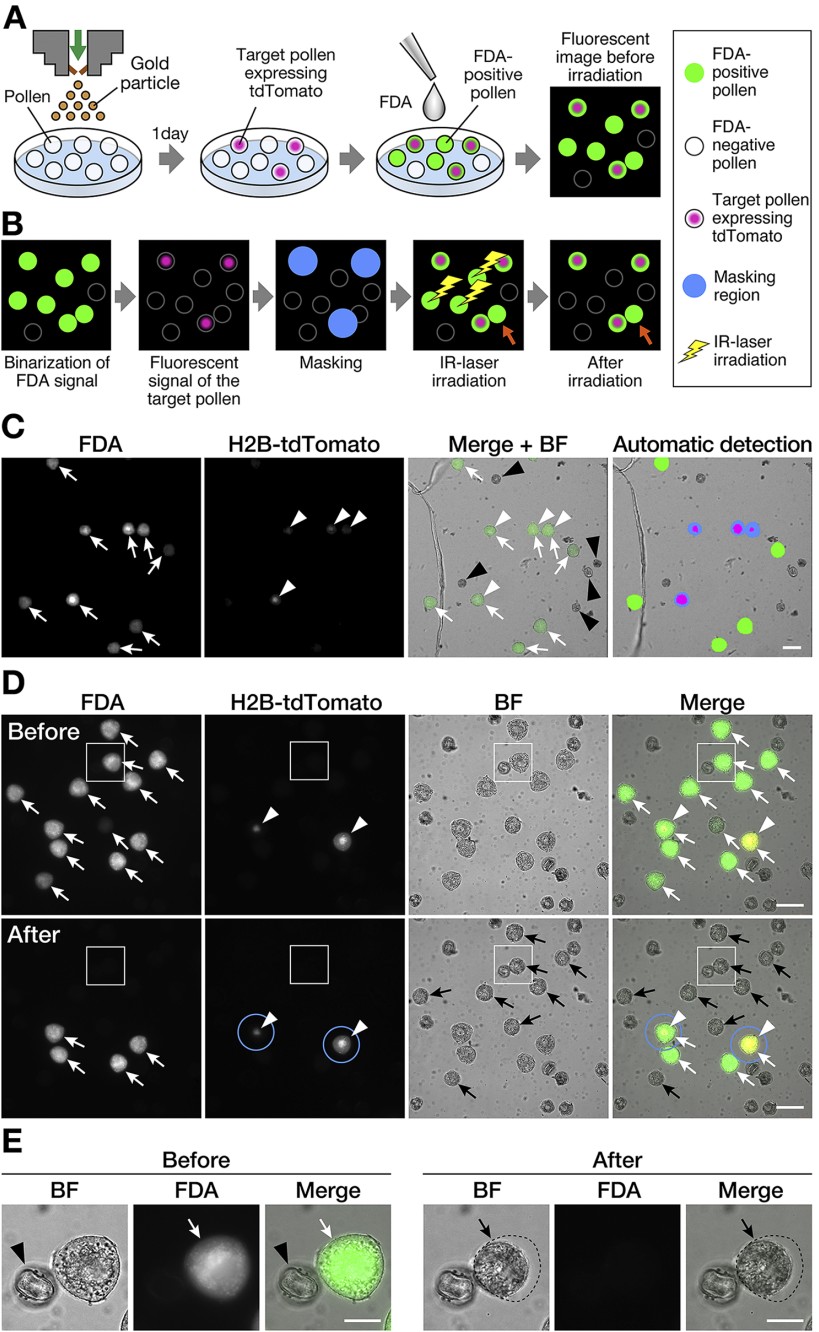

**Fig. 1.** Visualisation of target cells and laser irradiation of non-target cells. (a) Visualisation of target cells and non-target cells. Biolistic delivery of gold particles with plasmid DNA into the pollen was performed to visualise the target pollen. The nuclei of the plasmid DNA-introduced pollen were labelled using red fluorescent proteins (H2B-tdTomato). Fluorescein diacetate (FDA) solution was added to the medium immediately before laser irradiation. Non-fluorescent FDA absorbed by pollen is converted to the green fluorescent metabolite fluorescein, which is used as an indicator of viable pollen grains. (b) Automated cell disruption for target cell isolation by infrared (IR) laser irradiation. First, green and red fluorescence signals were detected through image capture before irradiation. FDA-positive pollen grains were defined and binarised by the captured image to detect a living cell. The target pollen expressing red fluorescent proteins were also detected by the captured image. To protect these target grains from laser irradiation, a diameter of 60 μm from the centre of the target cell was masked (blue-filled circle). Based on the *x, y* location of these defined pollen grains, IR laser irradiation was performed targeting the centre of each non-target pollen. After irradiation, the target cells were observed to maintain both FDA- and protein-derived fluorescence signals. Orange arrows indicate the non-target pollen attached to the target pollen, which was omitted from laser irradiation by masking. (c) Automated cell detection for target pollen isolation using programmable software. Fluorescence and brightfield images were captured for the detection of fluorescent signals and pollen shape. The detection program was applied to the images to automatically discriminate between non-target pollen with laser irradiation and tdTomato-expressing target pollen without laser irradiation. The detected area was automatically coloured with the following colours: green, FDA-positive pollen; magenta, tdTomato-expression; blue, masking area. (d and e) Brightfield and fluorescence images of pollen before and after IR laser irradiation. FDA-negative pollen grains were considered non-viable and were omitted from laser irradiation. The pollen with nuclei labelled with H2B-tdTomato were defined as target pollen for cell isolation. Blue solid line circles denote the masking diameter of 60 μm from the centre of the target cell. After laser irradiation, non-target pollen showed no FDA signal and shrinkage. The pollen close to the tdTomato-expressing target pollen in the masking circle was not irradiated, and thus, the FDA signal was retained. (e) Magnified images of the area are demarcated by the white-bordered square in (d). The shape of pollen before laser irradiation is shown as a black dotted-line circle. White arrow, FDA-positive pollen; white arrowhead, tdTomato-expression; black arrow, FDA-negative pollen; black arrowheads, FDA-negative shrunken pollen after laser disruption. Scale bars, (c and d) 50 μm and (e) 20 μm.

step program (including three automated detection, definition, and irradiation processes and two automated image capturing processes) was developed for the assessment of laser irradiation and consisted of the following steps: (a) image acquisition, (b) detection and definition of living pollen and its position on the microscope stage, (c) detection and masking of the target pollen, (d) IR laser irradiation of non-target pollen, and (e) image acquisition after irradiation (Figure 1a–c). In step a, the region of interest including the uniformly distributed pollen within an area of over 20 mm$^2$, was defined as the irradiation area. Thereafter, automatic image capture was executed based on green fluorescence signals, such as the FDA signal, and red fluorescence signal with respect to tdTomato expression. In step b, the FDA-positive pollen was detected automatically. To detect a living cell, a fluorescence signal intensity >9,600 was defined as the threshold of FDA positivity. Thereafter, binarised images were obtained using a general analysis task. An FDA signal corresponding to a diameter of >12 μm was defined as a single pollen to exclude dust. The positional information of each FDA signal was recorded using a region list task. Similarly, a fluorescence signal intensity >65,535 was defined as the threshold of the target cells based on fluorescent protein expression. In step c, an area with a diameter of 60 μm from the centre of the target cell was masked and not irradiated to protect the target cells from laser irradiation. In step d, based on the *x, y* location of these defined cells, moving to the centre of each non-target pollen, shutter opening/closing for irradiation was performed. Finally, the irradiated area was captured using tiling images to assess the laser disruption of the non-target pollen in step e. Steps b–d were executed in a loop that was automatically performed for each tile. To define the FDA-positive/negative signals in steps a and e, the average signal intensity was set as the threshold by randomly selecting 10 pollen grains that produced almost no fluorescence. All capture and irradiation processes were performed using an inverted microscope (Ti2-E) at 24–28°C under dark conditions.

### 2.8. Examination of laser conditions for single pollen disruption

Laser conditions for single-pollen disruption were examined. After suspending *N. benthamiana* pollen in the culture medium, a 1% (w/v) solution of FDA in acetone was immediately added at a final concentration of 0.019 μM. Pollen was irradiated using an IR laser. The laser power delivered to the samples at the focal plane was changed from 24 to 194 mW, which was measured using a power meter (Coherent). The laser source range was 50–400 mW. The duration of irradiation varied between 1 and 20 ms. After irradiation, the time for which the signal intensity of the FDA fluorescence was less than the threshold was measured at 5 min. The fluorescence did not disappear after 5 min and was undetectable. The average timing was obtained for two to three pollen samples.

### 2.9. Pollination of the disrupted pollen population under the semi-in vivo assay

A 1:1 mixture of *N. benthamiana* wild-type and transgenic *AtUBQ10p::tdTomato* pollen was used for the laser irradiation. A half of the anthers were used per laser irradiation and then pollinated on a pistil. Because the *N. benthamiana* pistil is too thin for semi-*in vivo* assays, the *N. tabacum* pistil was used instead for pollination. The FDA fluorescent dye was not used because it may disturb pollination, whereas pollen morphology, that is,

pollen shape under the bright field and pollen autofluorescence under the CFP bandpass filter (Nikon), were used as indicators of pollen grains. Unlike dust and other cells, *N. benthamiana* pollen shows strong blue autofluorescence, making pollen shape easy to detect. The tdTomato-expressing pollen was defined as the target pollen, and laser irradiation was omitted. A laser power of 145 mW at the focal plane and 10 ms irradiation time were used for the disruption of the wild-type pollen. After IR laser irradiation, the disrupted pollen bulk population was collected through the filter and pollinated to *N. tabacum* pistil (emasculated before 2 or 2 d). Immediately, the pollinated style was excised with a razor and placed horizontally on the pollen germination medium solidified with 1% (w/v) NuSieve GTG Agarose (Lonza, Switzerland), followed by incubation at 25–30°C for 18–24 hr under humid and dark conditions (Nagahara et al., 2021). Pollen tubes emerging from the cut end of the pistil were observed under an inverted fluorescence microscope (Eclipse Ti2-E; Nikon).

## 3. Results

### 3.1. Conditions of IR laser irradiation for selective disruption of non-target pollen

The FDA signal remained after 5 min of irradiation at 24 mW, even when the irradiation time was increased from 1 to 20 ms (Supplementary Table S1). In contrast, the FDA signal disappeared after irradiation at >48 mW. At an irradiation power of 48 mW, 165–225 s were required for the FDA fluorescence to disappear. The fluorescence signal disappeared within a short period of time after irradiation at >96 mW. At 194 mW, laser irradiation caused the pollen to jump and shift forward from the position observed before irradiation; such irradiation power was thus considered too strong for this study. For laser irradiation durations of 10 and 20 ms at 194 mW, the fluorescence disappeared simultaneously with laser irradiation (Supplementary Table S1); therefore, pollen was damaged during laser irradiation. To irradiate an increased number of non-target pollen grains without affecting the target cells, reducing the laser power and shortening the duration of irradiation is necessary to the maximum possible extent is necessary. Therefore, we determined the irradiation condition as the disappearance of fluorescence within 42.5 s, that is, a laser power of 145 mW at the focal plane and irradiation time of 10 ms, which was used for subsequent experiments.

### 3.2. Software-automated identification of pollen expressing exogenous genes is affected by pollen density and separation

First, the percentage of FDA-positive pollen in wild-type *N. benthamiana* was determined. As a result, 151 pollen grains were FDA-positive and 45 were FDA-negative, which indicated that 77.0% of wild-type *N. benthamiana* pollen grains were viable. Because a short laser irradiation period is preferred, FDA-negative pollen grains, which accounted for 23.0%, were excluded from subsequent laser irradiation. To identify the target pollen, a plasmid DNA vector containing the red fluorescent protein was introduced into the pollen. Gene expression was evaluated based on fluorescence signals at 18 hr after bombardment (Figure 1a). Each nucleus in the target pollen was labelled by expression of the introduced *AtUBQ10p::H2B-tdTomato* plasmid DNA. Five independent experiments on different days were used for the analysis (Table 1). Because it is necessary to irradiate many non-target

**Table 1.** Number and proportions of pollen before and after laser irradiation in the five samples

| | Irradiation area (mm²) | # of tiles covering area | # of the total pollen | Pollen (/mm²) | Before | | | After | | | Time irradiation (min) | Disruption rate %) | Viability pollen (%) |
|---|---|---|---|---|---|---|---|---|---|---|---|---|---|
| | | | | | FDA positive | tdTomato expressing | Target pollen (%) | FDA positive | tdTomato expressing | Target pollen (%) | | | |
| I | 22.5 | 80 | 1011[a] | 44.9 | 539 | 20 | 3.7[b] | 39 | 12 | 30.8[b] | 51 | 94.8[c] | 60.0[d] |
| II | 24.3 | 99 | 1617[a] | 66.5 | 638 | 35 | 5.5[b] | 185 | 31 | 16.8[b] | 68 | 74.5[c] | 88.6[d] |
| III | 26.3 | 100 | 2739[a] | 104.1 | 918 | 86 | 9.4[b] | 202 | 43 | 21.3[b] | 77 | 80.9[c] | 50.0[d] |
| IV | 55.2 | 120 | 2241[a] | 40.6 | 1432 | 94 | 6.6[b] | 396 | 62 | 15.7[b] | 83 | 75.0[c] | 66.0[d] |
| V | 51.7 | 117 | 3764[a] | 72.8 | 1457 | 53 | 3.6[b] | 462 | 31 | 6.7[b] | 63 | 69.3[c] | 58.5[d] |
| Average | | | | 65.8 ± 25.4[e] | | | 5.8 ± 2.4[e] | | | 18.2 ± 8.8[e] | 68.4 ± 12.4[e] | 78.9 ± 9.8[e] | 64.6 ± 14.6[e] |

[a] Pollen including both viable (FDA-positive) and non-viable (FDA-negative) pollen grains.
[b] The percentage of tdTomato expressing pollen out of FDA-positive pollen.
[c] The percentage of pollen that was irradiated by the laser and lost its FDA signal among non-target pollen.
[d] The percentage of tdTomato-expressing pollen that showed FDA fluorescence after laser irradiation.
[e] Average with standard deviation was shown.

pollen grains, the following five steps were executed and auto-matically performed by the graphically programmable software module: (a) image acquisition, (b) detection and definition of living pollen and their position on the microscope, (c) detection and masking of the target pollen, (d) IR laser irradiation of non-target pollen, and (e) image acquisition after irradiation (Figure 1b,c). The irradiation area is defined in step a. In step b, pollen viabil-ity was assessed by FDA staining. The FDA-positive pollen was considered viable and its position information was automatically registered. Pollen grains were suspended in pollen culture medium and stained with the FDA solution. After IR laser irradiation, the non-target pollen showed shrinkage and disappearance of the FDA signal (black arrows in Figure 1d,e), whereas the target pollen retained fluorescent signals derived from both FDA and H2B-tdTomato (Figure 1d). This indicated that the irradiated cells lost their viability. In contrast, non-target cells in the vicinity of the target cells (i.e., within the masked area) remained fluorescent because they were omitted from laser irradiation (Figure 1d). Of the five samples, 1,011–3,764 pollen grains, including 539–1,457 FDA-positive pollen grains, were counted (Table 1). The pollen density of each sample was 40.6–104.1 cells/mm$^2$ (average 65.8 ± 25.4 cells/mm$^2$; Table 1). Similarly, pollen showing a fluorescent signal derived from tdTomato expression was defined as the target cell for step c. The number of target pollens per experiment was 20–94, and the ratio of target pollen within viable pollen was 3.6%–9.4% (average 5.8 ± 2.4%, Table 1). Therefore, more than 90% of the pollen was defined as non-target pollen that needed to be disrupted by laser irradiation. Based on the *x, y* location of these defined cells, laser irradiation was performed at the centre of each non-target pollen in step d. Finally, the irradiated area was captured to assess the laser disruption of the non-target pollen. The average time required for laser irradiation was 68.4 ± 12.4 min for an irradiation area covering 80–120 tiles, when the average pollen density was 65.8 ± 25.4/mm$^2$ (Table 1). Sample I, which had the lowest pollen density, had the highest cell-isolation efficiency. The proportion of tdTomato-expressing pollen after irradiation was 8.3-fold higher, increasing from 3.7% to 30.8% (Table 1). These results indicate that pollen density affects isolation efficiency.

To investigate the effects of other factors on the isolation effi-ciency, we performed a detailed analysis of the irradiation results for each tile, including at least one tdTomato-expressing pollen. Laser irradiation was performed on samples with pollen grains of one-fourth or one-sixth anthers suspended over an area of 25 mm$^2$. First, the number of pollen grains in a single anther was calculated. The average pollen number in a single anther was 12,717 ± 2,814 (10 anthers from 5 flowers); thus, one-sixth of the anther was estimated to contain approximately 2,119 pollen grains, and the pollen density was calculated as 84.8/mm$^2$. Similarly, one-fourth of anthers were estimated to contain approximately 3,179 pollen grains, and pollen density was calculated as 127.2/mm$^2$. Next, the results for each tile were divided into two groups based on the presence or absence of pollen separation (Table 2). The number of pollen grains, pollen separation, and disruption success rate was counted for each tile (Table 2). When the pollen density was 84.8/mm$^2$, the average disruption success rates were 87.2 ± 13.4% and 71.6 ± 14.1%. This was higher than the 72.0 ± 22.7% and 65.7 ± 19.9% for pollen density of 127.2/mm$^2$. In contrast, when comparing the presence or absence of pollen separation, the disruption success rates were 87.2 ± 13.4% and 72.0 ± 22.7% for successful pollen separation, which were higher than that of unsuccessful pollen separation (71.6 ± 14.1% and 65.7 ± 19.9%). The maximum target pollen isolation rate 62.8 ± 35.5% and the disruption success rate

87.2 ± 13.4% were observed at the pollen density of 84.8 mm$^2$ with pollen separation. This result suggests that the isolation efficiency was affected by both pollen density and pollen separation, and the lower pollen density achieved more success.

### 3.3. Pollen germination and pollination of the isolated pollen

In the case of bombardment, pollen germination occurs before the expression of the transgene; therefore, pollen germination and pollination of isolated pollen cannot be studied (Nagahara et al., 2021). Transgenic *35Sp::H2B-tdTomato* was used for pollen germi-nation analysis, and *AtUBQ10p::tdTomato* pollen grains were used for pollination analyses as target pollen expressing fluorescent pro-teins. To analyse pollen germination of the isolated pollen, the wild-type and *35Sp::H2B-tdTomato* pollen grains were mixed, and pollen with nuclei labelled with tdTomato (white arrowheads in Figure 2) was defined as the target pollen and omitted from laser disruption. Pollen grains that did not show tdTomato fluorescence signals were disrupted using IR laser irradiation as a non-target pollen. After 1 hr of laser irradiation, germination of the target pollen grains with both the FDA signal and tdTomato expression was observed (white arrows in Figure 2), whereas it was not observed in the shrunken non-target pollen grains disrupted by laser irradiation (black arrows in Figure 2). The FDA fluorescent signals in several pollen grains disappeared, and these pollen grains showed bursts despite being protected from laser disruption (white dotted circles in Figure 2). Pollen tubes from the target pollen grains continued to elongate, and nuclei were transported to the pollen tube tips 3 hr after laser irradiation (yellow arrows in Figure 2).

Pollination and pollen tube growth of the isolated pollen were analysed in the pistil using a semi-*in vivo* growth assay (Bosch & Hepler, 2006). Wild-type and *AtUBQ10p::tdTomato* pollen grains were mixed at a 1:1 ratio. The tdTomato-expressing *AtUBQ10p::tdTomato* pollen grains were defined as target pollen and omitted from the laser disruption. Although pollen mor-phology and autofluorescence, but not FDA fluorescence, were used as indicators, pollen grain detection and laser irradiation were successfully performed (Supplementary Video 1). After laser irradiation, the pollen bulk population (including disrupted non-target pollen and tdTomato-expressing target pollen) was pollinated with *N. tabacum* pistil (Figure 3a). A semi-*in vivo* growth assay revealed that pollen tubes emerged from the cut end of the pistil. After pollination, the number of pollen tubes that emerged was counted using each pistil (Figure 3b,c). A clear difference between irradiated (+) and non-irradiated (−) pollen pollination was observed in the proportion of pollen tubes expressing tdTomato (Figure 3b). In the case of the non-irradiated pistil #C1, 57 pollen tubes were observed, including 36 wild-type pollen tubes and 21 tdTomato-expressing pollen tubes (Figure 3c). The percentage of tdTomato-expressing pollen tubes was 36.8%. In the case of the irradiated pistil #IR1, there were 52 pollen tubes, including 17 wild-type pollen tubes and 35 tdTomato-expressing pollen tubes. Thus, the percentage of tdTomato-expressing pollen tubes was 67.3% (Figure 3c). The average percentage of tdTomato-expressing pollen tubes in the three laser-irradiated pistils was 77.2 ± 8.6%, which was more than approximately two times that of the control experiment #C1 (Figure 3c). The percentage of tdTomato-expressing pollen tubes was higher in the laser-irradiated bulk pollen population, suggesting that disruption of non-target pollen was effective.

These results suggest selective target pollen germination *in vitro* and *in vivo* by cell disruption of non-target pollen.

**Table 2.** Number and proportions of pollen before and after laser irradiation in each tile

| Pollen density (/mm²)[a] | Separation of pollen[b] | Tile | Before | | | | After | | | | Disruption success rate (%)[d] | Average of disruption success rate (%)[e] |
| --- | --- | --- | --- | --- | --- | --- | --- | --- | --- | --- | --- | --- |
| | | | FDA positive | tdTomato expressing | Target pollen (%)[c] | Average target pollen (%) | FDA positive | tdTomato expressing | Target pollen (%)[c] | Average of target pollen (%) | | |
| 84.8 | True | 1 | 18 | 1 | 5.6 | 19.4 ± 15.3 | 3 | 1 | 33.3 | 62.8 ± 35.5 | 88.2 | 87.2 ± 13.4 |
| | | 2 | 14 | 2 | 14.3 | | 5 | 2 | 40.0 | | 75.0 | |
| | | 3 | 11 | 1 | 9.1 | | 3 | 1 | 33.3 | | 80.0 | |
| | | 4 | 10 | 1 | 10.0 | | 4 | 1 | 25.0 | | 66.7 | |
| | | 5 | 10 | 4 | 40.0 | | 4 | 4 | 100.0 | | 100.0 | |
| | | 6 | 9 | 1 | 11.1 | | 3 | 1 | 33.3 | | 75.0 | |
| | | 7 | 7 | 1 | 14.3 | | 1 | 1 | 100.0 | | 100.0 | |
| | | 8 | 5 | 1 | 20.0 | | 1 | 1 | 100.0 | | 100.0 | |
| | | 9 | 2 | 1 | 50.0 | | 1 | 1 | 100.0 | | 100.0 | |
| | False | 1 | 31 | 2 | 6.5 | 13.9 ± 7.7 | 9 | 0 | 0.0 | 31.8 ± 24.4 | 69.0 | 71.6 ± 14.1 |
| | | 2 | 25 | 4 | 16.0 | | 12 | 3 | 25.0 | | 57.1 | |
| | | 3 | 22 | 2 | 9.1 | | 6 | 2 | 33.3 | | 80.0 | |
| | | 4 | 21 | 1 | 4.8 | | 10 | 1 | 10.0 | | 55.0 | |
| | | 5 | 15 | 2 | 13.3 | | 7 | 1 | 14.3 | | 53.8 | |
| | | 6 | 15 | 3 | 20.0 | | 4 | 2 | 50.0 | | 83.3 | |
| | | 7 | 13 | 2 | 15.4 | | 3 | 2 | 66.7 | | 90.9 | |
| | | 8 | 10 | 3 | 30.0 | | 3 | 2 | 66.7 | | 85.7 | |
| | | 9 | 29 | 3 | 10.3 | | 10 | 2 | 20.0 | | 69.2 | |
| 127.2 | True | 1 | 26 | 2 | 7.7 | 15.1 ± 5.6 | 5 | 1 | 20.0 | 36.9 ± 28.3 | 83.3 | 72.0 ± 22.7 |
| | | 2 | 22 | 2 | 9.1 | | 15 | 3 | 20.0 | | 40.0 | |
| | | 3 | 14 | 3 | 21.4 | | 6 | 3 | 50.0 | | 72.7 | |
| | | 4 | 8 | 1 | 12.5 | | 0 | 0 | 0.0 | | 100.0 | |
| | | 5 | 8 | 1 | 12.5 | | 2 | 1 | 50.0 | | 85.7 | |
| | | 6 | 7 | 1 | 14.3 | | 3 | 1 | 33.3 | | 66.7 | |
| | | 7 | 6 | 1 | 16.7 | | 3 | 1 | 33.3 | | 60.0 | |
| | | 8 | 6 | 1 | 16.7 | | 4 | 1 | 25.0 | | 40.0 | |
| | | 9 | 4 | 1 | 25.0 | | 1 | 1 | 100.0 | | 100.0 | |
| | False | 1 | 13 | 1 | 7.7 | 11.2 ± 14.7 | 10 | 1 | 10.0 | 23.8 ± 28.8 | 25.0 | 65.7 ± 19.9 |
| | | 2 | 17 | 1 | 5.9 | | 5 | 1 | 20.0 | | 75.0 | |
| | | 3 | 15 | 1 | 6.7 | | 7 | 1 | 14.3 | | 57.1 | |
| | | 4 | 20 | 2 | 10.0 | | 7 | 1 | 14.3 | | 66.7 | |
| | | 5 | 46 | 1 | 2.2 | | 13 | 1 | 7.7 | | 73.3 | |
| | | 6 | 18 | 1 | 5.6 | | 6 | 1 | 16.7 | | 70.6 | |
| | | 7 | 25 | 2 | 8.0 | | 12 | 2 | 16.7 | | 56.5 | |
| | | 8 | 19 | 1 | 5.3 | | 7 | 1 | 14.3 | | 66.7 | |
| | | 9 | 2 | 1 | 50.0 | | 1 | 1 | 100.0 | | 100.0 | |

[a] The pollen including both viable and non-viable pollen grains was calculated by cell counter.
[b] Each pollen recognised separately as single pollen was identified true. The tile with pollen attached to each other as shown by arrows in Figure 1b was determined as false.
[c] The percentage of tdTomato expressing pollen out of FDA-positive pollen.
[d] The percentage of pollen that was irradiated by the laser and lost its FDA signal among non-target pollen.
[e] Average with standard deviation was shown.

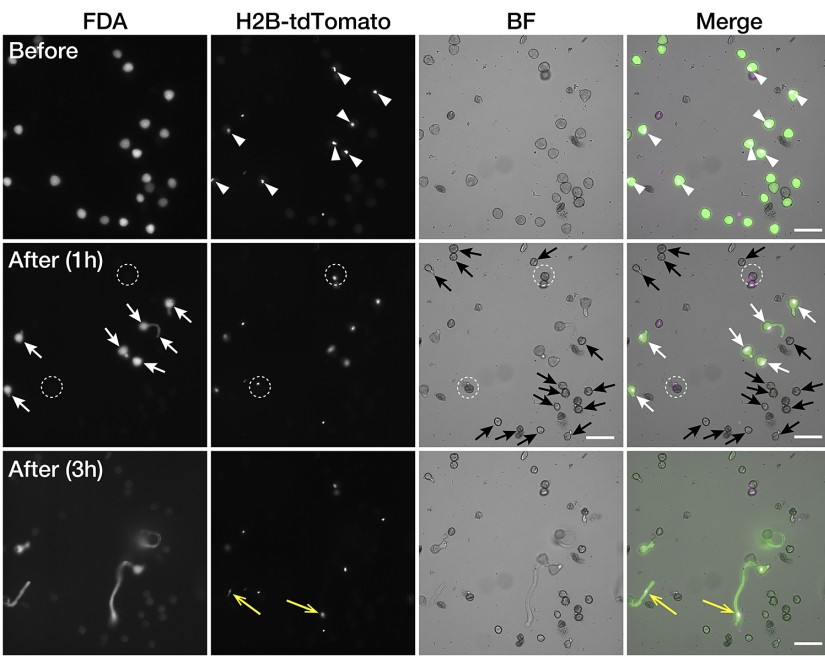

**Fig. 2.** Automated target pollen detection and germination of the isolated pollen Fluorescence and brightfield images of pollen before and after infrared (IR) laser irradiation. Pollen germination in the media after laser irradiation was observed 1 and 3 hr after laser irradiation. Pollen with H2B-tdTomato-labeled nuclei was defined as the target pollen for cell isolation (white arrowheads). Fluorescein diacetate (FDA)-positive pollen grains without tdTomato expression were defined as non-target pollen. After laser irradiation, pollen tube germination from the target pollen was observed (white arrows). Pollen grains that disappeared from the FDA signal did not show pollen germination (white dotted-line circles). Pollen tube nuclei transported to the pollen tube tips were observed 3 hr after the laser irradiation (yellow arrows). Scale bars, 100 μm.

## 4. Discussion

Distinguishing cells of interest from other cells is important to better understand biological phenomena. With recent developments in molecular cell biology, there is an increasing need to study gene functions and phenotypes of particular cells in multi-cellular organisms (including the study of cell lineages, cell fates, cell differentiation, and cell-to-cell communication). Here, an IR laser was used for the disruption of cells; ultraviolet or visible light can be used as long as the laser is focused on the focal plane. Laser light can damage cells through thermal, mechanical, and photochemical effects, which, in turn, elicit cellular stress and lead to cell death. Among these laser light sources, IR lasers are suitable for inducing gene expression in living organisms at the single-cell level. IR lasers have a high cell-heating capacity because they operate at a wavelength of 1,480 nm, which can heat water with a $10^5$-fold higher efficiency than 440-nm lasers can (Kamei et al., 2009). Therefore, IR lasers have been used to effectively induce gene expression using heat-shock promoters. In plants, only the female gametophyte in the ovule is stimulated and the heat-inducible Cre-loxP recombination system induces transgene expression (Hwang et al., 2019). In addition to inducing gene expression, IR lasers have been used in single-cell ablation and disruption experiments, because their high-power energy can be focused on a defined area. Laser disruption is used frequently in plant developmental studies. For example, laser ablation of the quiescent centre of *Arabidopsis* roots demonstrated that stem cells maintain their status by inhibiting the differentiation of the surrounding cells (van den Berg et al., 1995, 1997). Disrupted *Arabidopsis* embryo initial cells induce cell fate conversion of the suspensor cell to compensate for the embryo even after the cell fate has been specified (Gooh et al., 2015). These irradiation experiments have been performed on single cells in multicellular tissue but have rarely been applied for the selection of target cells. Cell damage depends on the IR-laser

power and duration of exposure; however, the effective conditions differ among cell species (Kawasumi-Kita et al., 2015). Therefore, we first identified a suitable irradiation setting for the disruption of non-target pollen. In the current study, a method for the automatic irradiation of non-target cells was established to enable efficient selection of target cells. Our method can be applied not only to pollen but also to other isolated cells such as spores, algae, and fungal cells. It does not require specific culturing areas or dish shapes as long as the cell population is the focus. To ensure effective selection, the cells and background, as well as the target and non-target cells, should be binarised precisely. Target cells were binarised using dyes and fluorescent proteins. The number of non-target living pollen grains was recorded using FDA staining, which is commonly used to assess pollen viability (Heslop-Harrison et al., 1984). FDA is hydrolysed by cytoplasmic esterase to produce fluorescein, which accumulates in the cell and exhibits a strong green fluorescence. In contrast, dead cells are characterised by the absence of esterase activity, and thus do not exhibit fluorescence. Generally, mature wild-type anthers contain a proportion of undeveloped or dead pollen (Lee et al., 2008). Additionally, pollen viability is reduced depending on storage and culture time (Hashida et al., 2012). In *Nicotiana*, anthers contain normal pollen as well as undeveloped, abnormally shrunken, or dead pollen (Matoušek et al., 2020). Thus, only living pollen was counted using FDA staining. In our experiments, 77.0% of the *N. benthamiana* pollen grains showed FDA signals, which was considered viable. After laser irradiation, the FDA fluorescent signals in several target pollens disappeared despite being protected from laser disruption (Figure 2). This may be because of arrest during pollen development for endogenous reasons, or bombardment and FDA solution may have negatively affected pollen development. In addition to FDA, other indicators, such as cell state and morphology, can also be used to define target cells (Supplementary Video 1). An appropriate density of cells is also important for effective isolation because a high cell density

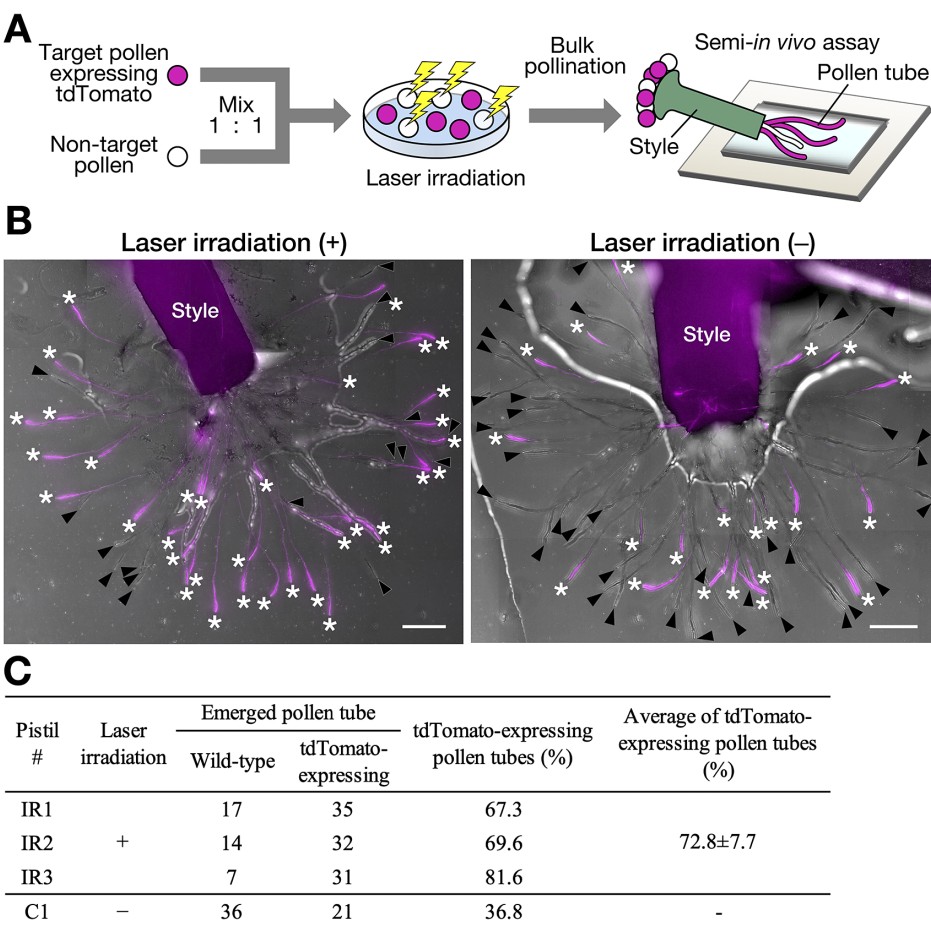

**Fig. 3.** Pollination of the isolated pollen (a) Schematic representation of the semi-*in vivo* growth assay using the isolated pollen. Target pollen expressing tdTomato and non-target wild-type pollen were mixed at a ratio of 1:1 and infrared (IR) laser irradiation was performed. After irradiation, the bulk pollen population was pollinated to the pistil, and pollen tubes emerging from the cut end of the style were observed. (b) Semi-*in vivo* growth assay with bulk pollen population with (+) and without (−) laser irradiation. Red fluorescent and brightfield images were merged. The tdTomato signal is indicated in magenta. Asterisks, tdTomato-expressing pollen tube; arrowheads, tdTomato-negative pollen tube. Scale bars, 400 μm. (c) The number of pollen tubes that emerged in the semi-*in vivo* growth assay with the bulk pollen population. The laser-irradiated bulk pollen population was pollinated on three pistils (IR1–3). The bulk pollen population, without laser irradiation, was also pollinated on one pistil (c1).

responsible for more non-target pollen may lead to an overlap with the masking area, which reduces isolation efficiency. Once the conditions for isolation have been established, high-throughput studies will be possible through automated isolation.

Antibiotic-based selection based on resistance genes has generally been used to eliminate non-target cells (Southern & Berg, 1982). Antibiotic and herbicide-resistance genes have been widely used as selectable markers in transgenic plants and cells (Breyer et al., 2014). However, it is difficult to use antibiotics in some organisms and cell species that are already resistant or show varying selectivities at different concentrations (Angenon et al., 1994). In addition, physiological responses to antibiotics can be altered by their use. Because our method does not use antibiotic-based selection, it can be applied to select cells of interest. For example, it may be effective in the study of antibiotic resistance and drug discovery in multidrug-resistant bacteria (Lage et al., 2018). Moreover, the positional information before irradiation is completely maintained after laser irradiation, and our method facilitates continuous analysis, such as live imaging, for tracing individual cells. The method can be applied to other approaches, such as the analysis of cell-to-cell communication between disrupted and non-disrupted cells in scattered cells, similar to the known positive and negative pollen density effects on pollen tube germination (Boavida & McCormick, 2007).

Pollen germination is possible not only in the media but also on the pistil. We have previously reported that bombarded pollen can elongate pollen tubes on the pistil and reach the ovule (Nagahara et al., 2021). Our results suggest that pollination with isolated pollen may achieve efficient seed production from the target pollen (Figure 3). However, the proportion of pollen tubes expressing tdTomato was not 100%, which may need further improvement. A detailed analysis of the isolation results for each tile revealed that the disruption success rate showed a wide deviation, indicating the importance of both pollen density and separation (Table 2). For example, combining methods to spread pollen grains in pieces more reliably may improve the isolation efficiency. A high separation efficiency can be achieved by separating each tile with a plate or microfluidic device and collecting cells from each tile. This method may facilitate high-throughput studies at the single-cell level, such as gene expression analysis and cell dynamic imaging, in various organisms.

## Acknowledgements

We sincerely thank Dr. Ueda and Dr. Kurihara for providing the plasmid vector and T. Shinagawa and E. Matsumoto for preparing the plant materials and performing the experiments. We also thank the members of the JST A-STEP project for helpful discussions.

**Open peer review.** To view the open peer review materials for this article, please visit http://doi.org/10.1017/qpb.2022.24.

**Financial support.** This work was supported by the Japan Science and Technology Agency (JST) [Adaptable and Seamless Technology Transfer Program through Target-driven R&D (A-STEP, JPMJTR194H)], Japan Society for the Promotion of Science [Grant-in-Aid for Transformative Research Areas (20H05778, 20H05779)], and Program for Promoting the Enhancement of Research Universities (2022). This work was also supported by Advanced Bioimaging Support in MEXT/JSPS KAKENHI (22H04926).

**Conflict of interest.** The authors declare that they have no conflict of interest.

**Authorship contributions.** Y.M. conceived and designed the study. I.K., M.I., and Y.M. collected data. Y.M., I.K., and T.H. wrote the manuscript.

**Data availability statement.** The data and processes of program that support the findings of this study are available from the corresponding author, Y.M., upon reasonable request.

**Supplementary Materials.** To view supplementary material for this article, please visit http://doi.org/10.1017/qpb.2022.24.

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
