## [Reviewer Report]

February 18, 2022

Prof. Olivier Hamant

Editor-in-Chief

Quantitative Plant Biology

Dear Prof. Olivier Hamant,

We would like ask you to consider our manuscript entitled “Target pollen isolation using automated infrared laser‒mediated cell ablation” for publication in Quantitative Plant Biology as an original article.

In the plant biological sciences, various imaging techniques have been used to study molecular and cellular dynamics for understanding biological phenomena. In this study, we aimed to establish a method for target cell isolation from the isolated cell populations using automated infrared laser-mediated cell ablation systems. After isolation, target cells were retained at the same location as before the treatment and were enriched in the cell population. This method can facilitate high-throughput studies at the single-cell level, such as gene expression analysis and quantification of cell dynamics, independent of any specific microfluidic devices, cell species and skilled professionals.

-All the authors have read and approved this manuscript.

-The manuscript has not been previously published, nor is it under consideration for publication elsewhere, including the Internet.

-There is no one who has reviewed this paper.

-To the best of our knowledge, there are no potential conflicts of interest to this study.

We look forward to seeing valuable comments from you and referees.

We will appreciate your consideration in reviewing our report.

Sincerely,

Yoko

---

Yoko Mizuta, Ph.D.

Institute of Transformative Bio-Molecules (WPI-ITbM),

Nagoya University,

Furo-cho, Chikusa-ku, Nagoya 464-8601, Japan

e-mail: mizuta.yoko.u6@f.mail.nagoya-u.ac.jp

---

## [Reviewer Report]

*Comments to Author*: This manuscript is reporting a scheme comprised of “H2B-tdTomato gene transfer through particle bombardment targeting N. benthamiana pollen on a glass-bottom dish”, “FDA staining,” “position register of the living transgenic pollen,” and “IR-laser ablation targeting the pollen of not-interest.” The setting conditions on IR-laser ablation were examined based on the binarized FDA signal intensities, indicating the frequencies of the ablated pollen of not-interest and the living transgenic pollen.

I understood that the reported scheme is feasible for live-cell imaging because the position is precisely maintained. However, the authors performed the precise position control with the originally equipped system, and this report is not on the development of the system itself. Although the ablation of pollen except for living transgenic one succeeded in this study, it isn't easy to imagine the practical experimental design in high-throughput and single-cell analyses.

It is critical to show an analytical example of an experiment using isolated pollen. The “isolated” pollen has remained on the glass-bottom dish. How are these living transgenic pollen moved from here and used? For example, are the ablated pollen population collected as bulk and available for the following analysis, such as pollination?

The technical novelty is hardly found in the current content. If a kind of analytical example using the “isolated” living transgenic pollen is indicated, the manuscript would be acceptable.

<Minor points>

(1) Some of the non-interest pollen escaped from ablation because such pollen is located in the vicinity of the living transgenic pollen. How about examining the masking region size (diameter) or adjusting the pollen density in a glass-bottom dish?

(2) The last sentence in section 3.1., “…140 mW laser power…” is 145 mW correctly?

(3) Second sentence from the last in section 3.2, “No pollen showing FDA-positive signals expressed red fluorescent protein” is correct?

(4) The words “bombarded pollen” (ex. Table 3) are a bit confusing because bombarded pollen does not always synthesize the transgene product, actually. How about using “tdTomato-positive pollen”?

---

## [Reviewer Report]

*Comments to Author*: In this study, the authors aimed to isolate target pollen by laser ablation. Usually, cell isolation was conducted by selecting target cells, but this has several disadvantages, such as damage to cells and low throughput. Cell isolation by ablation of non-target cells is reverse thinking, and it is safer than using the target cell method. Furthermore, implementing automatization turns the ablation system into a high-throughput method. Overall, I agree with the author’s claims in this study and the novelty of this study. However, while the manuscript is written understandably, there are sentences that are difficult to read or sometimes hard to grasp. I commented on several points as follows.

#1 Introduction

“The methods used to individually recognize and separate cells of interest from other cells facilitate studying their biology, physiology, biochemistry, and taxonomy at the single-cell level (Gross et al., 2015).”

Finally, I could recognize “facilitate” is the actual verb in this sentence (not “used”, “recognize”, and “separate”). It was really confusing to read this sentence. This is a typical example; I strongly recommend authors carefully check each sentence and improve them throughout the manuscript.

#2 Introduction

“Laser capture microdissection, which was recently combined with image analysis algorithms (computer-assisted microscopy isolation) (Brasko et al., 2018), facilitates capture of individual target cells. However, accurate definition is complicated when cells show complex shapes. Micromanipulation, such as manual cell picking with micropipettes using a microscope, is a simple approach for single-cell isolation; however, its throughput is limited, and skilled professionals are required. Additionally, the above methods are not suitable for sequential analysis to evaluate the effects of environmental factors under constant conditions because cell location and environment are altered during cell isolation.”

I finally recognized that these sentences are explained as category (1) after repeated reading of this manuscript.

First, I thought that the explanation of category (1) is completed in this sentence.

“Fluorescence- activated cell sorting (FACS) and microfluidic droplet-based platforms are methods of category (1), which have been widely used for high-throughput isolation; however, this requires a large number of cells, and these harvesting techniques interfere with the cell location and phenotypic profile.”

And then the explanation of category (2) started after several sentences.

“Regarding methods of category (2), probably the most commonly…”

I was confused about what the intermediate sentences are (Laser capture microdissection…).

I strongly recommend the authors to mention these sentences belong to category (1).

#3 Introduction

In this manuscript, the authors developed a high-throughput ablation system with Nicotiana benthamiana pollen. However, there was almost no explanation about why the authors chose pollen and Nicotiana benthamiana in this study in the introduction section. Thus, I could not sufficiently catch the significance of this study from the introduction section. Part of these explanations seems to be mentioned in the discussion section. I think it is better to move these sentences to the introduction.

#4 Pollen vs. Pollens vs. Pollen grains

The grammatically correct plural of “pollen” is pollens. It is not largely used. If you refer to pollen in plural, but you do not want to use “pollens” then you can use “pollen grains”. Check this throughout the manuscript.

For example, page 12: “In this study, the pollen from one anther were suspended in pollen culture medium and were stained using FDA solution.”

#5 section 3.1

In this study, pollen viability was only defined by the FDA signal. It is possible that IR-irradiation disappears the FDA signal but pollen still may be viable. If the authors confirm pollen viability after irradiation by other evidence (such as by pollen germination ratio), it will support the authors claim. Or, it is better to cite it if the authors observed apparent morphological change after irradiation (for example, bursting pollen or leakage of pollen content, etc.). Even though the authors observed no morphological difference, it is important to cite them in the text.

In section 3.4

“The non- target pollen showed no FDA signal and shrank, whereas the target pollen retained both fluorescent signals derived from FDA and H2B-tdTomato (Figure 2).”

If this “shrank” indicate shrunk pollen grain, I recommend to explain it in more detail with a figure containing microscopic observations.

In the discussion, the authors wrote as follows,

“Not only the FDA, but other indicators such as cell state and morphology can also be used. “

This information should be placed in the result section and should be directly shown by evidence with a figure.

#6

Related to #5, it is important to show pollen viability about successfully bombarded pollen. I thought this was one of the important results of this study. (i.e., Isolated pollen is available (or not available?) for a subsequent experiment and making progeny.)

#7 section 3.2

“Pollen occur as isolated cells when collected from the anther, which is a simple collection method.”

This was a very confusing sentence for me.

#8 section 3.2

“After bombardment, gene expression was evaluated based on expression of fluorescent proteins at 18 hab (Figure 1).”

If the abbreviation “hab” is brought back to formal wording, it becomes as follows.

“After bombardment, gene expression was evaluated based on expression of fluorescent proteins at 18 h after bombardment (Figure 1).”

I feel strange about two “after bombardment” appearing in a single sentence. Is it OK for you?

#9 section 3.2

“A total of 20−94 introduced pollen grains were obtained per bombardment (the average efficiency of gene introduction was 57.6 ± 31.9 pollen).”

I could not understand why authors cite the average value here. What is “average efficiency”? I feel this is redundant because 57.6 ± 31.9 is 25.7-89.5, and these values are close to 20-94 and felt that this is meaningless. The authors should remove it or explain it in more detail.

I was also confused about reading this sentence.

“3.6%−9.4% (mean 5.8 ± 2.4%, Table 2).”

What does “mean” mean? Different statistic was applied with “average efficiency”?

#10 Fig1

No explanation for white circles in A1-3 and black circles A4. I assumed that they are aborted pollen (or non-stained pollen?). It should be explained in the Figure.

#11 Fig2

I think it is better to put symbols to ablated pollen grains in Figure2 “after”. I assumed that authors want to show this system can ablate non-target pollen efficiently. Currently, only failed pollen grains were emphasized by arrows.

#12 Page 5 – Plant materials and growth conditions

“Two-weeks-old seedlings were transferred to soil and were grown in a green room at 25–30 °C under long-day conditions (16 h light/8 h dark) in a greenhouse at 25–30 °C.”

Here the authors mention the green room and greenhouse. Is there any difference? If yes, please explain these differences or remove one.

#13 Page 12

“In step (1), the region in interest within the field of view was defined.”

It should be “the region of interest”.

#14 Discussion

“This indicates that the fluorescent signal derived from the FDA of approximately one third of the pollen was below the threshold. Target bombarded pollen is masked and omitted from laser irradiation, the cause of which may be photobleaching of the FDA dye to under the threshold. The fluorescent proteins and dyes used for target cell visualization should thus be photostable. This method may facilitate high-throughput studies at the single- cell level, such as gene expression analysis and cell dynamic imaging in various organisms.”

I could not understand what the authors would like to claim here.

Do these sentences imply that FDA is not suitable for cell isolation?

---

## [Reviewer Report]

*Comments to Author*: Dear Dr. Mizuta and colleagues,

We have just received the comments of two expert reviewers on your manuscript submission. Their detailed comments are attached.

As you will see, a reviewer suggests the changes that could be addressed by rephrasing and modulating the figure. But the other reviewer asks to add the data showing the analytical example using the isolated pollen. 

It would be ideal if actual experimental data could be added. But if this is not possible, I think it would help if you could propose several examples of a feasible analysis.

We would be happy to receive a corrected version of your manuscript when it is ready. 

I thank you again for having submitted your excellent manuscript to Quantitative Plant Biology.

Best regards,

Minako Ueda

---

## [Reviewer Report]

*Comments to Author*: The revised manuscript has been improved by showing the germination ability of pollen isolated by IR-laser ablation. In addition, overall descriptions have been more understandable than in the previous version. I agree with these improved points; however, the present content seems not to meet the level for publication yet. This is because the structure of the content is still hard to understand, and the use of inappropriate values is shown to some extent. I strongly recommend being subjected to English editing by the main expert. If the authors want to claim the feasibility of this study, more quantitative data is required. Also, it is recommended to reconsider the claim possible from the present results.

In Fig. 2B, a higher proportion (67.3%) of pollen tubes germinated from the target pollen were indeed observed; however, at least 3 biological repeats should be performed to obtain the quantitative data statistically. Moreover, the proportion is less than 2 times that of the control experiment (36.8%). Although the enrichment of target pollen succeeded, it seems not enough as the feasible level for the future analyses raised by the authors.

The authors added the reason why they chose Nicotiana pollen for this study and the possible application in “Introduction” in response to reviewer 2. I agree with this point because the result in Fig. 2B coincides with this claim. Therefore, please revise the “Discussion” section in accordance with the “Introduction” section. If authors insist on “high-throughput studies at single-cell level” as the future application, a specific example should be raised. What can be evaluated by using bombarded and isolated pollen, even if the introduced gene is not expressed before pollen tube germination?

The critical point of this study is how to increase the purity (rate of content) of the target pollen among the samples after the laser ablation loops. That’s why I requested to evaluate the density as one of the previous comments. I meant that pollen density can be adjusted quantitatively by calculation because the pollen is suspended with a liquid medium before being mounted on the glass-bottom dish.

In the revised manuscript, why did the authors use UV excitation (for CFP image) for indication of the pollen grains? I wonder it because pollen viability seemed to be easily judged by the morphology in bright field. Please add the explanation.

<Other specific points>

<b>Section 3.1</b>

The value 165-190s (line 3 in this section) should be revised to 165-225s. The authors finally set the irradiation time at 10 ms. Please add the reason why 20 ms was not employed in spite of the required time for irradiation is shorter than that at 10 ms.

<b>Table 1<b>

● Please add a caption explaining “N. D..”

● Please move the item words “Laser power at the focal plane (mW)” to the upper side of the values.

● How about rephrasing the title “Time required for disappearance of FDA signals in irradiated pollen.”

<b>Section 3.2 and Table 2<b>

● Regarding Table 2 and the part after the descriptions from “Thereafter, the number of non-target pollen……”, the numbers of FDA-positive pollen in the population from a single anther (ca. 20000 pollen grains) are indicated? Also, I wonder why the values as the number of FDA-positive pollen is the product of the multiplication of “Irradiation area” by “Density.” Does it mean that all cells mounted on a glass-bottom dish were FDA-positive? Please confirm if these values are appropriate.

● In the text, the authors mentioned the frequency and the average. It would be helpful if you add these items to Table 2.

<b>Section 3.3<b>

● The values in the sentence “Using multiple fields of view……” (lines 12-15 from the start of this section) are not included in both Tables 2 and 3. Please indicate the source and process to obtain these values. If this description corresponds to my question regarding FDA-positive pollen in Table2, this part should be moved to section 3.2.

● At line 18 in the same section, the values 24.0-28.2 should be revised to 24.0-34.9.

● Overall, it is difficult to understand the correlation between the “Irradiation area,” “The xy tiling,” and “fields of view.” Similarly, the mixture use of “processing” and “irradiation” is confusing. Please clarify the definition of these words, or try to unify the word with the same meaning. The item words in tables also should be minded.

● In the last part of this section, the authors described the time required for a single loop. Please add this information in Table 2. I couldn’t calculate these values from the present information. It would be helpful for readers to indicate the source information.

● In the same part, the author expressed the time using “minutes.” Please unify the expression in the text and Table 2.

<b>Section 3.4<b>

● Indeed the result described in the title was obtained, but the correlation between the FDA-positive pollen density and the target isolation efficiency was not seen in the indicated results. I understand this type of study tends to show wider deviation. It might be good to mention the range of the obtained efficiency, in addition to the descriptions of the best result.

● As for another point, the present descriptions focus only on the FDA-positive pollen density. I think the overall density, including both viable and non-viable pollen, influences the efficiency more because it influences the effectiveness of masking. Therefore I’ve requested the evaluation of pollen density. Moreover, it seems better to combine the present Tables 2 and 3.

<b>Section 3.5<b>

● The 35Sp::H2B-tdTomato transgenic tobacco was not mentioned in the materials and methods. Please add the related information.

● Regarding the sentence “The FDA fluorescent signals in several pollens disappeared, and these pollen grains showed bursts despite being kept from laser ablation” (from line 12th in this section), please add discussion. What is considered to cause this? And I recommend replacing “pollens” with “pollen grains.”

● From lines 18 to 19, which transgenic line was used (35Sp::H2B-tdTomato or AtUBQ10p::tdTomato)?

<b>Fig. 2A<b>

Please indicate the cyan circle lines also on the H2B-tdTomato picture after the irradiation.

<b>Legend of Sup. Video 1<b>

The size of the masking area is described as “30 µm”, but “60 µm” in the legend of Fig. 1. Please revise to the correct value.

---

## [Reviewer Report]

*Comments to Author*: In this study, the authors aimed to isolate the target pollen by laser ablation. This version of the manuscript has really improved readability with additional data about pollen tube elongation in a culture medium and semi-in vivo assay. I think that the current version of the manuscript provides enough novelty and data to be accepted. I put some comments to improve readability as follows.

#1 Low selectivity for target pollen grains.

The authors claimed increased selectivity of target pollen as follows(page 17),

“The highest cell isolation efficiency occurred in sample I, and the proportion of tdTomato-positive pollen after irradiation was 8.3-fold higher, increasing from 3.7 % to 30.8 %.”

For me, the ratio (30.8%-3.7%, Table 3) of targeted pollen/total pollen does not seems to be a high selectivity to me.

In this point, the authors stated (page20-21),

“The appropriate density of cells is also important for effective isolation because a high cell density responsible for more non-target pollen may lead to an overlap with the masking area, which reduces isolation efficiency (Figure 2).”

But I thought alternative idea of improving the selectivity ratio would be possible. For example,

ex1. Improving area definition

Now it seems if non-target pollen overlapped with the masking region, it will be not ablated; however, improving ablation condition by imaging program(or other method?), I think selectivity will be drastically improving.

ex2. Two times ablation after physically moving pollens

If the addition of one more ablation step after (gently) shaking the media after the 1st ablation, the selectivity ratio will be improved.

I recommend the authors discuss the improvement of the selectivity ratio.

#2 Table 1

#2-1

The column “laser power at…” should be located above the mW number (i.e., “24”)

#2-2

What “N.D.” means? I could not find any explanation for this abbreviation.

(“No data”? If it is yes, all columns of 24mW are unnecessary, “Not disappeared”? If so, the authors should explain it).

#3 Table 3

“Viability of bombarded pollen (%)” may not be correct.

“Viability of tdTomato-positive pollen before and after IR-laser irradiation (%)” may be correct.

#4 Fig1B

From this figure, this method seems to be able to select target pollen with 100% selectivity. However, the current method is far from 100% (max. 30.8%). I recommend putting at least one FDA positive pollen (non-target pollen) in “After irradiation” figure to avoid overestimating/misunderstanding for the readers.

#5 Fig 2B

I am really confused after reading this sentence (page 18).

“The tdTomato- expressing pollen tubes were 35 in number, i.e., 67.3 %, (out of a total of 52 tubes that underwent laser irradiation) (Figure 2B), whereas there were 21 out of a total of 57 pollen tubes that did not undergo laser-irradiation (36.8 %, Figure 2B).”

#5-1

Please distinguish Figure 2B legend using such as “2B left/right” or “2B -Laser irradiation/+Laser irradiation”.

#5-2

I also could not catch where the numbers “35, 52, 21, 57” came from.

I recommend author cite this data somewhere (in Fig2B?, making a new plot or table?).

#5-3

I could not understand why tdTomato signal was colored green. First, I misunderstood this green signal as FDA. I recommend that the authors should use a uniform color code throughout the manuscript. But if it is difficult technically, I can accept this figure.

#6

“Pollen grains obtained from flower buds were used for biolistic delivery of plasmid vectors into cells”

I think that it would increase the reproducibility if the authors could state in which developmental stage the flower buds were.

#7

“We previously reported that bombardment pollen…”

It should be “We previously reported that bombarded pollen…”

---

## [Reviewer Report]

*Comments to Author*: Dear Dr. Mizuta and colleagues,

We have just received the comments of two expert reviewers on your revised manuscript. Their detailed comments are attached.

As you will see, a reviewer requested the evaluation of pollen density and the biological repeats of pollen tube germination experiment. I think both of these are feasible and would be very important for publication in Quantitative Plant Biology. 

We would be happy to receive a corrected version of your manuscript when it is ready. 

I thank you again for having submitted your excellent manuscript to Quantitative Plant Biology.

Best regards,

Minako Ueda

---

## [Reviewer Report]

Dear Editor,

We wish to thank you and the peer reviewers for very positive reviews and for very useful and insightful comments regarding how to improve the manuscript. We are sorry that our response and edits are so late because we had to conduct additional experiments. Now, I am pleased to inform you that all of the requests have been met, and we not only highlight places in the text where edits were made, but also provide a detailed response to peers’ requests below.

All authors have seen and assisted with the editing of the revised manuscript and approve the re-submission to Quantitative Plant Biology. There are no conflicts of interest.

Please feel free to let us know if any requests remain and we look forward to seeing or findings published in Quantitative Plant Biology.

Sincerely,

Yoko

---

## [Reviewer Report]

*Comments to Author*: The revised manuscript contains newly added quantitative data enough to claim the future feasibility of the reported scheme. The present data satisfies the level for publication in QPB. In minding reader-friendly construction, it might be better to revise the order of descriptions in sections 3.2 and 3.3 because some sentences are repeated.

1) It seems good to combine sections 3.2 and 3.3. My suggestions are below;

>Move the sentences at lines 1-6 from the bottom of page 14 (First, 151 pollen grains……., were excluded from subsequent laser irradiation.) to the top of section 3.2.

>Remove the sentences from line 20 of p13 to line 9 of p14 (The FDA solution was added at 18h, ………Using the above five samples, we isolated the target pollen by cell disruption of non-target pollen with IR laser irradiation.).

>Move and place the sentences from lines 2-7 of p16 after the sentence “Pollen grains were suspended in pollen culture medium and stained with the FDA solution” (p14).

>Replace the sentence “As the number of pollen grains increased……Irradiation time depended on the area and number of pollen grains.” (lines12-14 in p15) with the sentence “Sample I, which had the lowest pollen density……These results indicate that pollen density affects isolation efficiency” (lines 17-21 in p16).

>Remove the remained part in section 3.3, retitling the combined section.

2) Other minor points;

>Section 3.3 and Table 2; Authors evaluated the separation of pollen, but I couldn’t well imagine the state of “pollen separation” from the caption. In the case of “True”, does it means that all pollen grains in a single tile are independently dispersed without merging or contacting each other? It would be helpful for readers if more detailed explanations were added.

>Legend “a” in Table 2: culturated -> calculated

>Line 6 of p17: How about adding “(10 anthers from 5 flowers)” after “12,717 ± 2,814”.

>Lines 20-21 of p21: How about if you mention the actual counted data of pollen FDA-staining?

>Lines 10-12 (p13) and Table S1: Although it could be my misunderstanding, 1 minute is 60,000 milliseconds. If the unit in Table S1 is ms, all the indicated times are less than 1 minute. Please confirm the unit for time.

---

## [Reviewer Report]

*Comments to Author*: This version of the manuscript has improved tables (i.e., clear and informative) and improved figures (i.e., intuitive and close to results). Furthermore, the authors added some tables, which are quantitative and well-support the data. The authors replied to my comments properly.

---

## [Reviewer Report]

*Comments to Author*: We are pleased to inform you that your manuscript has finally been accepted. We would appreciate it if you correct the minor points pointed out by Reviewer1.

---

## [Reviewer Report]

Dear Editor,

It is a great pleasure to accept our manuscript.

We corrected the minor points pointed out by Reviewer 2 in the revised manuscript R3 as a final version.

We look forward to seeing publishing in Quantitative Plant Biology.

Sincerely,

Yoko